# Relationships between Resilience and Self-Efficacy in the Prosocial Behavior of Chilean Elementary School Teachers

**DOI:** 10.3390/bs14080678

**Published:** 2024-08-05

**Authors:** Sonia Salvo-Garrido, Karina Polanco-Levicán, Sergio Dominguez-Lara, Manuel Mieres-Chacaltana, José Luis Gálvez-Nieto

**Affiliations:** 1Departamento de Matemática y Estadística, Universidad de La Frontera, Temuco 4780000, Chile; 2Programa de Doctorado en Ciencias Sociales, Universidad de La Frontera, Temuco 4780000, Chile; k.polanco01@ufromail.cl; 3Instituto de Investigación FCCTP, Universidad de San Martín de Porres, Lima 15102, Peru; sdominguezl@usmp.pe; 4Departamento de Diversidad y Educación Intercultural, Universidad Católica de Temuco, Temuco 4780000, Chile; mieres@uct.cl; 5Departamento de Trabajo Social, Universidad de La Frontera, Temuco 4780000, Chile; jose.galvez@ufrontera.cl

**Keywords:** prosocial behavior, self-efficacy, resilience, teachers, elementary school

## Abstract

Teachers’ actions go beyond instruction, as their personal traits influence their teaching methods, problem-solving skills, and the quality of their relationships with students. Among these attributes, their prosocial competencies stand out for contributing to school, community, and social coexistence. Furthermore, the connection they have to resilience and self-efficacy promotes increased effectiveness in meeting the demands of an ever-more challenging work environment. This research aimed to analyze the effect of the relationship between self-efficacy and resilience on the prosocial behavior of Chilean elementary school teachers. The sample consisted of 1426 teachers (77.2% women) working in public and subsidized Chilean schools. Structural equation modeling (SEM) explored the relationships between self-efficacy, resilience, and prosocial behavior. The findings indicate that self-efficacy and resilience directly and positively affect the prosocial behavior of elementary school teachers. It is suggested that resilience, self-efficacy, and prosociality among teachers are promoted due to their synergistic effects and, consequently, the benefits for school children, especially those from vulnerable social contexts.

## 1. Introduction

Teachers play a crucial role in students’ cognitive, social, and emotional development [1], with schools being vital institutions for their socialization during childhood and adolescence [2,3]. In stressful and demanding environments [1,4], teachers face daily challenges, with actions impacting individuals, communities, and society, recognizing that teachers’ behaviors have implications at individual, community, and societal levels [5]. Prosociality, valued by society [6], acts as a protective factor for teachers in challenging situations [7], positively influencing children who interact with them by receiving their values, affection, and altruism [6,8,9].

There are essential competencies for teachers that go beyond mere academic instruction, especially when working with children, given the challenging nature of their work [10,11,12]. Teachers’ socioemotional skills are crucial for establishing effective relationships in the classroom and strengthening the bond with students [13,14], enhancing the teaching and learning process, especially in contexts of high socioeconomic segregation like Chile, where an impact on academic performance and student development is observed [15,16,17,18]. In a constantly changing world, adaptation is crucial [4,19,20,21,22], highlighting the importance of resilience, self-efficacy, and prosocial behavior for teachers to achieve their pedagogical goals and foster the comprehensive development of their students [3,23].

An analysis of the correlation between resilience, self-efficacy, and prosocial behavior is crucial because of its substantial impact on well-being and educational quality. Although previous research has established this connection in other contexts [3,24], the focus on teachers is particularly pertinent. As mentioned in previous paragraphs, teachers play a crucial role in shaping students’ socio-emotional skills, and understanding how their own personal characteristics influence their prosocial behavior can provide valuable insights for improving the educational environment. Furthermore, self-efficacy and resilience are essential skills that empower teachers to confront challenges and adjust to demanding educational settings, both personally and professionally. These skills, in turn, can foster prosocial conduct in the classroom and benefit students.

Prosociality is deemed essential in teachers as it protects against adverse conditions, reduces burnout levels [7], and increases job commitment [25]. This concept has been defined as “dispositions, voluntary behaviors, and processes that focus on or contribute to the well-being of others” at different levels [26]. According to Caprara et al. [27], an individual’s prosociality is reflected in various actions such as helping, sharing (or comforting), caring, and feeling empathy. Currently, multiple studies have linked prosociality with empathy [28,29], helping behaviors, and the possibility of sharing [30], all very relevant in professional education and for practicing teachers [31,32]. Prosocial behaviors foster positive social interactions, benefiting both the giver and the receiver and leading to mutual support and social recognition for positively evaluated actions [6].

In this regard, the close teacher–student relationship fosters students’ prosocial behavior over time [25,33,34], enhancing adaptation, reducing exclusion [35] and bullying among students [36], as well as decreasing discriminatory and racist behaviors among peers [37]. It is crucial that children develop prosocial behavior [38], and thus, it is expected that teachers support the development of social competencies in their students, especially prosociality [39,40]. The bond between the student and the teacher allows children and adolescents to internalize values such as respect and affection in their interactions, which will influence prosocial behavior [8,9]. Moreover, students with attentional and behavioral difficulties improve their behavior management and peer relationships [41].

The perception of self-efficacy is crucial for teachers, enabling them to face challenging environments [17,42]. It is defined as judgments about one’s capabilities to achieve positive outcomes, influencing behavior, thoughts, and emotions [43,44]. This is evident in stressful and changing contexts [44,45,46], affecting effort and persistence in achieving goals [44]. Specifically, teacher self-efficacy is associated with the belief in the ability to facilitate student learning and achieve expected academic performance [46,47,48,49], strengthening the trust relationship with students [50] and promoting inclusive education [51,52].

Tschannen-Moran and Woolfolk Hoy [48] propose that teacher self-efficacy comprises different domains. The first is called Efficacy in Instructional Strategies, which refers to the perception of a teacher’s ability to generate learning strategies that meet the needs of the class. The second is Efficacy for Classroom Management, which reflects the teacher’s ability to support their students’ emotional and behavioral regulation to conform to classroom norms. The third is Efficacy for Student Engagement, which addresses the teacher’s performance related to the evaluation of their students’ own confidence and appraisal of their abilities, manifesting in students engaging properly in their activities. 

Resilience is a dynamic process that utilizes personal, school, social, and community resources to overcome challenges and adapt [53,54,55,56]. Adverse experiences can lead to learning, such as increased self-acceptance, the ability to face challenges, and greater compassion and connection with others, resulting in a beneficial internal transformation for others [57]. According to Saavedra and Villalta [58], resilience is reflected in how problems are approached, one’s self-perception, and personal beliefs, being built over time through experiences that provide continuity to personal development. 

Resilience enables individuals to face challenging and problematic situations by dynamically mobilizing resources of various kinds to solve the difficulties they experience [55]. Thus, resilience is not solely a personal, relational, and collective process. Therefore, the conditions present in the school context can function as obstacles or enhancers of these processes [22,59,60,61]. Additionally, teachers’ resilience fosters resilience in their students, which is crucial, especially in vulnerable contexts [53,62]. Teachers act as role models for their students due to the substantial amount of time they spend together, particularly during the early years of elementary education [63].

### 1.1. Relationship between Resilience and Prosociality

Resilience has been associated with prosocial behavior [3,64,65]; given the potential for personal growth that can arise from overcoming adversity, it is possible that adverse experiences encountered by individuals may foster resilience and strengthen social connections [57,66]. In this context, it is observed that teachers with greater resilience show better emotion regulation and greater empathy, implying that they evaluate the emotions of others more appropriately [56]. Moreover, resilience is negatively associated with stress, emotional exhaustion, and difficulties in maintaining discipline in the classroom [4,5], favoring the possibility of being available to help and support students [30].

Thus, according to Sunbul and Gordesli [23], resilience significantly predicts teacher prosociality. It enables coping with negative feelings linked to perceived obstacles or difficulties and increases the disposition to confront work-related issues constructively. Resilient teachers have a greater capacity to help other people voluntarily. The dynamic and prosocial classroom environment has a reciprocal effect on teacher resilience [67], as it serves as a required resource for the development of teacher resilience and also contributes to teachers’ social enjoyment [68,69,70,71].

### 1.2. Relationship between Self-Efficacy and Prosociality

Self-efficacy significantly predicts prosocial behaviors in people of different ages, especially in teachers [3,72,73,74,75]. The beliefs in one’s own self-efficacy influence the capacity for self-regulation, manifested in cognitive, motivational, affective, and behavioral aspects [43], facilitating the display of prosocial behaviors [27]. In this way, individuals are required first to perceive that they have the capabilities to face a situation and its consequences to provide help and support to others, making it important to consider the context and distinguish the required skills [6]. Specifically, caring for another person requires perceiving that they need help and what type of help they need [6].

Consequently, self-efficacy beliefs are linked to prosocial behavior, considering that the confidence to act prosocially is present [72,76,77]. The teachers with greater self-efficacy show a higher development of socio-emotional competencies [78] and higher emotional intelligence [79], and it is associated with the experience of feeling joy, pride, and love in interactions with others [80]. Meanwhile, the perception of teacher self-efficacy influences the exchange of knowledge among teachers and the prosocial behavior that increases the willingness to share useful resources and solve problems [81]. At the same time, teachers who provide socio-emotional and instructional support display prosocial behavior and cope better with problems influencing their students, fostering higher levels of self-efficacy and prosocial behaviors [82,83,84]. Prosociality in the classroom favorably promotes teachers’ professional self-efficacy beliefs and their own mental health [85]. It enables them to benefit from their positive interactions with their colleagues and students [67]. Reciprocally, the students strengthen their academic self-concept [86], express greater confidence [87], and increase their commitment, motivation, and sense of belonging [88]. This way, a better teacher–student relationship is encouraged, conflicts are reduced [89,90,91,92,93], and the management of students’ behavior benefits [94].

### 1.3. Relationship between Teacher Self-Efficacy, Resilience, and Prosociality

Recent research has explored how teachers’ self-efficacy and resilience are interrelated and affect their prosocial behavior. Developing personal competencies among teachers within the school context promotes prosocial participation; specifically, self-efficacy and resilience maintain a direct, positive, and significant relationship with prosocial behaviors [23]. It is important to mention that resilience and self-efficacy as personal characteristics are linked with contextual factors and are reflected in people’s behavior, i.e., they are expressed in prosocial behavior [3,23]. The interplay among these three variables also applies to peer interactions, since prosociality, resilience, and creative self-efficacy have a positive effect on adolescent students, specifically in reducing stress and increasing their well-being [95]. It is relevant to consider this, given the potential beneficial effect of the school and teachers on promoting such attitudes and behaviors [95].

Moreover, it has been observed that resilience positively relates to teacher self-efficacy [96,97,98]. Therefore, increased self-efficacy and resilience favor coordination with other teachers and collaborative work [99]. This is based on the fact that self-efficacy is relevant for facing adverse situations, bearing in mind the effort involved in achieving the proposed goals by overcoming obstacles that may arise along the way [44,80]; similarly, resilience facilitates the use of different types of resources that the person has when facing problems. Thus, designing and implementing professional development programs that increase self-efficacy and resilience can have a positive impact on promoting prosocial behaviors among students [42]. 

According to Gratacós et al. [96], resilience is a significant factor for teachers to enhance their adaptability in resolving adverse situations, which entails improving their self-efficacy and augmenting personal resources to confront obstacles better and alleviate adverse conditions [100]. The perception of self-efficacy is crucial for teachers to attain any set goal involving motivation, difficulty assessment, and decision making [101]. The individual’s belief in their capabilities and likelihood of success is more pertinent than their actual skill development [43,45,101]. This, combined with personal values, influences the inclination towards supportive behavior [6]. Consequently, prosocial behavior, self-efficacy, and resilience assist in navigating challenges in teaching [23,67]. Specifically, increased self-efficacy and resilience among teachers are associated with heightened prosocial behavior, thus benefiting their students.

Resilient people are better able to handle abrupt and negative changes, according to a study conducted on Italian participants aged 18 to 60 years. This is related to their self-efficacy beliefs, which support the adaptation of the person to withstand and recover from difficult circumstances and show greater positivity towards life. Resilient people with higher self-efficacy tend to manage their emotions better and, therefore, often express prosocial behavior [102]. The relevance of these three constructs and their connection (self-efficacy, resilience, and prosocial behavior) has been observed in adolescents as they favor their mental health [95]. 

It is worth noting that self-efficacy and resilience are separate constructs that can be measured independently; however, the scientific literature evidences that they are interrelated. This interrelationship is crucial to understanding how these personal competencies influence teachers’ prosocial behavior.

The conceptual framework for this study is presented in Figure 1.

Therefore, this research aims to analyze the effect of self-efficacy and resilience on prosocial behavior in Chilean elementary school teachers. The hypotheses are that self-efficacy and resilience have a direct and positive effect on teachers’ prosocial behavior (H1) and that resilience and self-efficacy are positively related (H2).

## 2. Materials and Methods

### 2.1. Participants

The study population consists of 85,298 Chilean teachers working in primary education, teaching students aged 6 to 13 years. These teachers are employed in municipal and subsidized schools. A random sample was estimated using the following strata: region, residence (urban, rural), type of education, and gender. The stratified multistage probability sampling was estimated with a 95% confidence level, a sampling error of 2.5%, and a variance of *p* = *q* = 0.5 [103]. Consequently, the sample comprised 1441 teachers (77% women, 22.6% men, and 0.4% with no information) with an average age of 41.5 years (SD = 10.8). Most schools were located in urban areas (81.2%), with 83.6% being public schools and 16.4% being subsidized schools. On the other hand, teaching experience measured in years ranged from less than one year to 48 years, with an average of 14.2 years (SD = 10.1). In the sample in this study, the term “teachers” refers exclusively to classroom teachers. Administrators and teachers of specialized subjects were not included. This ensures that the results focus on teachers with direct and continuous interaction with students in the classroom.

### 2.2. Instruments

The research instruments in this study were divided into two parts. The first part consisted of a sociodemographic questionnaire to capture information about the teacher’s age, gender, ethnic group, type of school, years of experience, and work sector, among others. The second part focused on gathering information related to three perception scales of self-efficacy, resilience, and prosociality, which are detailed as follows:

Teacher Self-Efficacy Scale (TSES). Originally developed by Tschannen-Moran and Woolfolk Hoy [48] to assess perceptions of teacher self-efficacy. The original proposal consists of 24 items answered on a five-point ordinal scale (1 = nothing, 5 = a lot), which evaluates three factors with eight items each: efficacy for instructional strategies (e.g., “To what extent can you ask good questions for your students?”), self-efficacy in classroom management (e.g., “What can you do to make children follow the classroom rules?”), and self-efficacy in student engagement (e.g., “How much can you do to help your students value learning?”). The results obtained in a psychometric study aimed at analyzing the evidence of validity and reliability in a population of Chilean elementary school teachers from public and private subsidized schools reveal an internal three-factor structure, as indicated in the original proposal, but with a general factor and three residual factors, and showed good psychometric properties (RMSEA = 0.069; 90%CI [0.065, 0.072]; SRMR = 0.015; CFI = 0.988; TLI = 0.981) and high reliability of the general factor in relation to the scores (α = 0.972) and to the construct (ω = 0.985) [104]. 

SV-RES60 Resilience Scale for Youth and Adults. This scale was constructed and validated in the Chilean population [58]. This instrument consists of 60 items measured on a Likert-type scale with 5 categories (1 = Strongly disagree, 5 = Strongly agree). It has 12 factors: Identity (I am/basic condition), Autonomy (I am/self-view), Satisfaction (I am/problem view), Pragmatism (I am/resilient response), Bonds (I have/basic condition), Networks (I have/self-view), Models (I have/problem view), Goals (I have/response), Affectivity (I can/basic condition), Self-efficacy (I can/self-view), Learning (I can/problem view), and Generativity (I can/respond). Regarding psychometric properties, it reports an adequate level of reliability, with a Cronbach’s alpha of 0.96, and adequate validity, with a Pearson linear correlation coefficient of 0.76 [58]. The results found in recent research on elementary school teachers conducted by Salvo-Garrido et al. [105] provide favorable evidence of validity and indicate a general factor and 12 residual factors, showing good psychometric properties (RMSEA = 0.032; 90%CI [0.030, 0.033]; SRMR = 0.012; CFI = 0.986; TLI = 0.977), as well as a high reliability of the general factor for the scores (α = 0.981) and for the construct (ω = 0.991). This suggests that the scale has good internal consistency and dependably measures the resilience construct.

Adult Prosocialness Behavior Scale (APBS). This scale was proposed by Caprara et al. [27] to assess prosocial behavior. It consists of 16 items scaled in a Likert format with 5 categories (1 = Never, 5 = Always) that are associated with the theoretical dimensions that compose prosocial behavior: helping (for example, “I am pleased to help my friends/colleagues in their activities”), sharing (for example, “I am willing to put my knowledge and skills at the disposal of others”), caring (for example, “I try to be close to and take care of those in need”), and empathizing (for example, “I empathize with those who need it”). This scale was adapted and validated with Chilean university students in teaching programs by Mieres-Chacaltana et al. [106], reporting a structure comprised of one general latent factor and four specific residual factors (helping, sharing, caring, and empathy), with adequate reliability of the general factor (α = 0.932; ω = 0.968) and a good level of fit (RMSEA = 0.042; IC90% [0.036–0.049]; SRMR = 0.012; CFI = 0.995; TLI = 0.988).

### 2.3. Procedure

Initially, all principals of public schools, mayors, and directors of local education services were contacted, considering that Chilean public schools are under the administration of municipalities and local education services depend on the Ministry of Education of Chile. The researchers presented the study to the relevant authorities to invite participation and obtain authorization to apply the instruments. Subsequently, those schools whose directors decided to participate in the research were sent a link containing the informed consent, the sociodemographic questionnaire, the Teacher Self-Efficacy Scale (TSES), the Prosocialness Behavior Scale (APBS), and the Resilience Scale (SV-RES60). The relevant directors then provided the information to the teachers. It is important to mention that the teachers who agreed to participate were informed about the ethical principles of the research, such as the voluntariness of participation, the right to withdraw at any time without prejudice to them, and the risks and benefits, among others. The data were collected on an online platform (Question Pro). Moreover, it is highlighted that this study has the approval of the Scientific Ethics Committee of the Universidad de La Frontera, Chile (Evaluation File No. 053_21; Study Protocol Sheet No. 019/21).

### 2.4. Data Analysis

This study adopted a structural equation modeling (SEM) approach to explore the relationships between self-efficacy, resilience, and prosocial behavior, as proposed in Figure 1. Data from 1406 elementary education teachers were collected and analyzed. A preliminary descriptive analysis was made of the univariate normality of the items by means of their skewness (<2) [107] and kurtosis (<7) [107]. As for the estimation method, the weighted least squares means and variance adjusted (WLSMV) method [108] was used, as recommended for analyzing ordinal variables [109] across a wide range of sample sizes [110]. Furthermore, WLSMV does not make distributional assumptions about the observed variables [111] and makes accurate estimates even if the data are biased [112]. Given the sensitivity of the Chi-Square coefficient to large samples, as is the case in this study, from an interpretative perspective, the model presents a suitable fit when the CFI and the TLI display values over 0.90 [113]; RMSEA values below 0.08 are considered adequate [114]; and SRMR below 0.08 [115] is also regarded as adequate. Regarding the influence assessment, low (<0.30), moderate (between 0.30 and 0.50), and high magnitude (>0.50) [116] were considered. Mplus v. 8.4 software was used [117].

## 3. Results

Table 1 displays the main descriptive indicators for the items by scale. Skewness was negative for all items on the three scales, indicating that scores were preferred towards the two highest categories. Unlike the other two scales, the kurtosis values for the SV-RES60 scale items were all positive. Thus, in general, skewness and kurtosis reached adequate magnitudes.

The results of the analysis, based on the total sample, show that the model evaluating the impact of self-efficacy and resilience on prosocial behavior in Chilean elementary school teachers (Figure 1) fits the data well [χ2 (4847, *N* = 1426) = 21,975.474, *p* < 0.000, χ2/df = 4.5338, with TLI = 0.911, CFI = 0.912, RMSEA = 0.050, CI (90%): 0.049–0.050, SRMR = 0.069].

Table 2 summarizes the standardized factor loadings of the observed items on the latent constructs. All were greater than 0.5 and statistically significant (*p* < 0.0001). Furthermore, 88% achieved values greater than or equal to 0.71.

The estimated parameter values of the theoretical model of the impact of self-efficacy and resilience on prosocial behavior shown in Figure 1 were 0.308 (β1) and 0.261 (β2). Both values were statistically significant (*p* < 0.001) and empirically supported the first hypothesis. Self-efficacy was a strong predictor of prosocial behavior over resilience, which was also important. The results showed a significant positive relationship between the variables of self-efficacy and resilience, with a correlation coefficient of 0.366 (*p* < 0.001). This supports the hypothesis that self-efficacy and resilience are interrelated, which is consistent with previous studies that have found that confidence in one’s own abilities is associated with personal resilience.

## 4. Discussion and Conclusions

This study represents an initial effort to apply an integrated model of self-efficacy theory and resilience theory to understand teachers’ prosocial behavior, considering the relevance of all mentioned concepts for research in the field of education. It also supports interventions that consider that self-efficacy, resilience, and prosocial behavior are fundamental variables for teachers to carry out successful work, achieve adequate performance in their students, and promote the comprehensive development of children. Despite this, most of the existing literature only considers these factors separately [62,65,78,84,91], thus failing to make a stronger evidence-based connection between the constructs and how these variables can predict prosocial behavior. Consequently, this study represents a new perspective considering individual self-efficacy and resilience predictors of prosocial behavior.

This study aimed to analyze the effect of self-efficacy and resilience on teachers’ prosocial behavior. The results confirm the initial hypotheses, underscoring the need to promote these competencies to benefit the school environment. It should be noted that self-efficacy and resilience positively influence teachers’ prosocial behavior, H1, as indicated by Sunbul and Gordesli [23]. These personal competencies enable overcoming obstacles and adversities [44,80], fostering prosocial behaviors [3,6,23,101]. This strengthens teacher collaboration [99] and improves student relationships [3,8,9,64]. 

Another significant finding in this research is that self-efficacy is a strong predictor of prosocial behavior, being a more relevant explanatory variable than resilience. The perception of self-efficacy is linked to an individual’s self-regulation, which fosters the display of prosocial behaviors, thereby facilitating action as a result of confidence in one’s abilities and the perception of less stress, which, depending on their values, will lead to the manifestation of prosocial behaviors in their interpersonal relationships [27,43,101]. Values enable individuals to transcend their concerns to pursue others’ well-being [6]. Consequently, self-efficacy is a fundamental personal resource for resolving difficult situations [100], which is necessary to provide care and support to others in need [6]. 

Regarding the second hypothesis, which states that resilience and self-efficacy are positively related, it is evident that both variables are positively and significantly associated in line with different research findings [96,97,98]. Resilience is crucial in teachers’ performance as it allows them to face difficult situations through personal, community, and school resources [53,54,55,56]. Additionally, teachers’ perception of self-efficacy enables them to perform in complex environments or face difficulties given their confidence in their own abilities in various areas that would allow them to successfully achieve their purposes [44,45,46,49]. Consequently, this fosters a better learning environment for children and the development of life skills that enable them to face and solve future situations [83]. 

The significance of this research lies in the positive impact of teachers’ prosocial behaviors on the educational community and society as a whole, a concept supported by [6]. As students observe their teachers throughout the day, they absorb academic content and learn valuable lessons on problem-solving and interpersonal relationships. Therefore, interventions aimed at promoting teachers’ prosocial behavior, encompassing actions like helping, sharing, caring, and empathizing, are crucial, considering the role of self-efficacy and resilience in fostering such behaviors [6,23]. Moreover, interventions focusing on teachers’ personal development could yield multiple benefits, including enhancing their perception of school support, strengthening collaborative networks within the educational community, and fostering constructive responses to professional challenges. Ultimately, these efforts would provide a more enriching educational experience for students, serving as a valuable asset for teachers in addressing the varied needs of their students and fostering a nurturing environment crucial for the success of inclusive schools [41].

The results of this study have important implications for teacher support and development. There are several ways to improve teacher self-efficacy and resilience, such as strengths-based coaching, peer mentoring, and a teacher assessment system that incorporates self-identified goals and strengths [118,119]. In addition, there are related programs that have proven effective, such as Social Emotional Learning for Adults, SEL [120], the Cultivating Awareness and Resilience in Education, CARE [13] program, and the ACHIEVER Resilience Curriculum [121]. The findings of this study can be valuable for designing interventions aimed at enhancing teachers’ skills, particularly in the areas of resilience, self-efficacy, and prosocial behavior. This will provide a better understanding of the interconnections among these skills. 

In conclusion, the findings of this study underscore the importance of self-efficacy and resilience in promoting prosocial behavior among elementary school teachers in Chile. Self-efficacy, as the strongest predictor, suggests that confidence in one’s own abilities is crucial to developing prosocial behaviors. On the other hand, resilience also plays a significant role, facilitating teachers’ ability to cope with adversities and support their students effectively. This study and its results highlight the relevance of fostering these personal competencies to improve school dynamics and students’ educational experiences. In addition, they can guide the development of training and support programs for teachers to enhance their competencies and the well-being of the educational community. They can also provide valuable information for school administrators and educational policymakers.

In terms of limitations, it is important to note that private school teachers were not included, which could limit the generalizability of the results to all types of schools in Chile. In addition, as this was a cross-sectional cohort study, temporal connections cannot be inferred. Future studies could consider a more diverse sample that includes teachers from private schools and different educational levels, such as preschool and high school. Furthermore, it would be valuable to explore how some additional factors, such as burnout, mental health, moral values, and social capital (community, family, school, peers), influence teachers’ prosocial behavior. It is also considered relevant that further research should focus on exploring how teachers’ self-efficacy, resilience, and prosociality may influence the effectiveness of inclusive schools.

## Figures and Tables

**Figure 1 behavsci-14-00678-f001:**
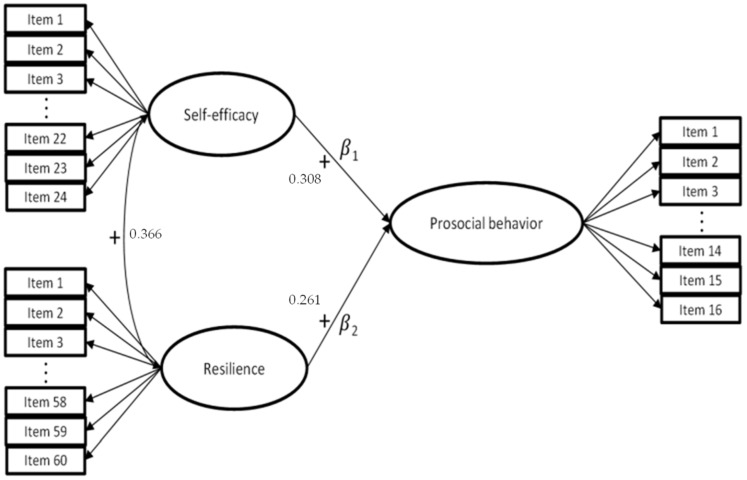
Theoretical and adjusted model of the impact of self-efficacy and resilience on prosocial behavior.

**Table 1 behavsci-14-00678-t001:** Descriptive statistics based on items by scale.

	Mean Itens			SD			g1		g2	
Scale	Min	Max	Total	Min	Max	Total	Min	Max	Min	Max
Self-Efficacy	3.78	4.34	4.07	0.76	0.96	0.84	−0.36	−0.92	−0.03	−0.57
Resilience	3.99	4.65	4.41	0.7	1.05	0.81	−1.01	−2.81	0.63	9.61
Prosocial behavior	3.29	4.71	4.16	0.7	1.16	0.88	−0.17	−1.75	−0.03	−3.3

Notes. Standard deviation (SD), skewness (g1), kurtosis (g2).

**Table 2 behavsci-14-00678-t002:** Standardized factor loadings of observed variables on latent constructs.

Scale	[0.570–0.697]	[0.710–0.798]	[0.801–0.898]	Total
Self-Efficacy		4	20	24
Resilience	9	24	27	60
Prosocial behavior	3	9	4	16
Total	12	37	51	100

## Data Availability

The data that support the findings of this study are not available because they are confidential data.

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
