# Peer review of "Relationships between Resilience and Self-Efficacy in the Prosocial Behavior of Chilean Elementary School Teachers"

_behavsci, 2024, doi:10.3390/bs14080678_

Round 1

Reviewer 1 Report

Comments and Suggestions for Authors

An interesting topic, especially because it focuses on teachers' personal characteristics that relate to wellbeing and educational quality. In general, the manuscript is well written. Authors could improve the theoretical part to promote justifiction and importance of reserach design and methodological choices for the manipulation of the variables. In other words, why is it important to check the relation between resilience and self-efficacy with prosocial behavior? Other research but not regarding teachers has somehow established this relation. Moreover, although one of the hypothesis is the connection between resilience and self-efficacy, it is not highlighted if this relation plays some role in prosocial behavior. For the variables a Bifactor-ESEM model seemed to be used to check factorial structure. However, to perform Bifactor-ESEM model a theoretical issue of a general structure should have been identified. Especially with resilience, prior research has found that different aspects of resilience relate differently to prosocial behavior. The model evaluating the impact of self-efficacy and resilience on prosocial behavior in Chilean primary school teachers has a moderate fit. It seems that the manipulation of the variables does not tell the whole story. The authors comment of the sample size. The model fit could be tested with a subset of the sample to check for sample size constraints. The discussion should include a more deep insight into the findings, and implications about practice and research could be provided. In some cases, sentences should be more clear and focused on the core concepts and scope of the manuscript. See for expample the first sentence of the abstact or the sentence: The bond between the student and the teacher allows children and adolescents to internalize values such as respect and affection in their interactions, which will influence prosocial behavior [8,9]. Moreover, students with attentional and behavioral difficulties improve their behavior management and peer relationships. In the title of the manuscript the primary grade should be referred and taken more into account in the theoretical part.

Comments on the Quality of English Language

Editing could improve quality of the language used. For example, relation/connection/ could be used instead of relationship for the link between resilience, self-efficacy and prosocial behavior.

Author Response

Dear revisor,

Thank you for your comments, which have helped us improve the manuscript. Various enhancements have been made to the article, highlighted in yellow. Attached is the response letter.

Reviewer 2 Report

Comments and Suggestions for Authors

The authors report on the relationships among teacher-perceived self-efficacy, resilience, and prosocial behavior in primary schools in Chile. The results of structural equation modeling (SEM) revealed the positive effects of self-acceptance and resilience on prosocial behavior. These findings can provide valuable information for school administrators and education policymakers, particularly in the context of teacher shortages, high turnover rates, and the potential for academic and behavioral problems leading to school failure.

The manuscript is well-written. Only several minor revisions are suggested below to further develop it.

P4, lines 169-178: It would be beneficial if it were clearly stated whether the term "teachers" in the sample refers to classroom teachers exclusively or includes administrators/specialized subject teachers. If those participants are included, it would be helpful for readers to know the proportions of each group.

P5, line 205: As described by the authors in the factors of resilience, self-efficacy can contribute to resilience, although they are still separate constructs and can be measured independently. It is suggested to briefly mention this type of relationship in section 1.3, "Relationship between Teacher Self-efficacy, Resilience, and Prosociality."

P5, line 208: In the sentence "..., with a Pearson linear correlation coefficient of 0.76," it is understandable that the Cronbach's alpha was calculated for internal consistency. How was the Pearson correlation used to assess reliability?

P7, lines 282-287: It would be great to include the results for the relationship between the self-efficacy and resilience variables.

Please check statistical symbols that need to be italicized (e.g., p, N, …).

It is suggested that the Discussion section include more implications of the study findings. For instance, there are a variety of teacher supports in improving their self-efficacy and resilience. These could include asset-based coaching, peer mentorship, an improved teacher evaluation system that incorporates self-identified goals and strengths, and more. Additionally, there are programs related to these supports such as Adult Social-emotional Learning (SEL), Awareness and Resilience in Education (CARE), ACHIEVER Resilience Curriculum, and practices to improve students' social behavior outcomes. The findings can be discussed in relation to the previously studied effectiveness of these programs.

Author Response

(The authors gave the same response as above.)

Reviewer 3 Report

Comments and Suggestions for Authors

Relationships between Resilience and Self-Efficacy in the Pro-social Behaviour of Chilean Teachers

This paper addresses the relationship between self-efficacy, resilience and pro-social behaviour.
The reference to a solid base of relevant documents provided a good foundation for the literature review. In addition, the diversity of sources and the inclusion of psychometric and mixed methods studies allowed for a comprehensive and grounded approach in the field of educational psychology.
The use of internationally validated scales in the study provides the study with some solidity.
Suggestions for improvement:
The authors should talk about the validity and reliability of the scales used to collect information. Not all readers are familiar with these scales and they could be better explained.
The analysis is quite comprehensive in relation to issues related to self-efficacy, resilience and the social behaviour of teachers in different educational contexts. However, I think there was room for further exploration of the problem of inclusive schools.
I feel there is a need for a concluding section or final considerations, where the authors situate the research context and clearly respond to the research objective.

Author Response

(The authors gave the same response as above.)

Reviewer 4 Report

Comments and Suggestions for Authors

The article need some imprivement.

Introduction

1. The authors should reference previous research on several key topics to provide a comprehensive background for this study. First, studies exploring the relationship between resilience and prosociality highlight how individuals' ability to recover from adversity can influence their propensity to engage in prosocial behaviors. Second, examining the link between self-efficacy and prosociality sheds light on how confidence in one's abilities impacts their willingness to help others. Finally, understanding the interplay between teacher self-efficacy, resilience, and prosociality can offer insights into how educators' beliefs in their capabilities and their resilience affect their interactions with students and their promotion of prosocial behavior in the classroom.

2. The aim of this research is not stated; please add it at the end of the introduction.

Literature review

Point 1.1: The second paragraph needs to include additional previous research concerning the relationship between resilience and prosociality. This should also be applied to points 1.2 and 1.3.

Method

In data analysis, what kind of statistical software is used for SEM? Please specify the software. Additionally, which authors use WLSMV, MLR, or ULSMV? Please clarify what authors use.

Results

- Please add the results of the figure based on the SEM analysis.

- In Table 1, which statistics are used? What are the minimum and maximum values for g1 and g2? Are there any missing outputs? If the authors used SPSS, are there results for skewness and kurtosis (statistic and standard error)? Please reanalyze. Also, remember that the decimal in English is a dot (.).

- Add the SEM outcome figure to clarify the meaning of the paragraph under Table 2.

- There is no section on limitations and future research. Additionally, there is no conclusion. Please add these sections.

Author Response

(The authors gave the same response as above.)

Round 2

Reviewer 4 Report

Comments and Suggestions for Authors

This manuscript meets the standards of this journal. I personally endorse the acceptance of this manuscript.